# Uncertainty Analysis in SAR Sea Surface Wind Speed Retrieval through C-Band Geophysical Model Functions Inversion

**Fabio Michele Rana *** and **Maria Adamo**

National Research Council of Italy (CNR), Institute of Atmospheric Pollution (IIA), c/o Department of Physics, University of Bari, Via Amendola 173, 70126 Bari, Italy; adamo@iia.cnr.it
* Correspondence: fabiomichele.rana@iia.cnr.it; Tel.: +39-0805442387 or +39-3804114171

**Abstract:** The purpose of the study is to assess the suitability of synthetic aperture radar (SAR) data to provide sea surface wind (SSW) fields along with a spatial distribution of both SSW speed and direction uncertainty. A simple methodology based on geophysical model function (GMF) inversion to obtain a spatial distribution of both SSW speed and its uncertainty is proposed. Exploiting a dataset of Sentinel-1 images, a sensitivity analysis of the SSW speed uncertainty is carried out on both the uncertainties and the mean values of SAR normalised radar cross section (NRCS), incidence angle and SSW direction, at different spatial scales. The results show that SSW speed uncertainty significantly increases with wind vector cell (WVC) dimension. Moreover, the dominant contribution to the SSW speed uncertainty due to both NRCS and SSW direction uncertainty must always be taken into account. A better precision and accuracy in the estimation of SSW speed and its uncertainty is evidenced by C-band model 7 (CMOD7) GMF rather than the C-band model 5.N (CMOD5.N). To evaluate the results of SSW retrievals, wind data from the European Centre for Medium-Range Weather Forecasts (ECMWF) model are also exploited for comparisons. Findings indicate a high correlation between the uncertainty from SAR estimations and that from the comparison of SAR vs. ECMWF.

**Keywords:** synthetic aperture radar (SAR); Sentinel-1; sea surface wind (SSW); geophysical model function (GMF); wind speed uncertainty

## 1. Introduction

Sea surface wind (SSW) is a physical essential climate variable (ECV), which crucially contributes to the characterization of Earth's climate system and its changes [1]. Specifically, SSW speed and direction represents an atmosphere surface ECV, and it belongs to the current list of ECVs as specified in the Implementation Plan for the Global Observing System for Climate (GCOS IP 2016), in support of the United Nations Framework Convention on Climate Change (UNFCCC) and the Intergovernmental Panel on Climate Change (IPCC).

SSW measurements may be provided by several different sources, such as in situ stations or buoys, e.g., from the United States National Oceanic and Atmospheric Administration (NOAA) National Data Buoy Center (NDBC), spaceborne active microwave sensors, like scatterometers, e.g., the advanced scatterometer (ASCAT) [2,3], or radar altimeters, e.g., the Joint Altimetry Satellite Oceanography Network (JASON) [4,5], and numerical weather prediction (NWP) models, e.g., the global European Centre for Medium-Range Weather Forecasts (ECMWF) and the regional Weather Research and Forecasting (WRF). Each source has pros and cons, especially in terms of temporal and spatial (or punctual) distribution, as well as precision and accuracy of wind measurements.

In situ stations or buoys are unfortunately costly and sparse, and not deployable in hardly accessible areas as well, although they can routinely collect high precision and high accuracy wind observations. They are typically used for in situ validation of wind estimates derived from other techniques and sources. On the contrary, satellite scatterometers

and radar altimeters are suitable for synoptic and mesoscale research on the ocean wind. Scatterometers provide very accurate winds in rain-free conditions with a typical wind speed error of about 1 m/s [6], at spatial grids up to 12.5 km (e.g., ASCAT coastal wind) [7]. Radar altimeters show instead a typical error of wind speed of about 1.5 m/s [5,8], however, providing as scatterometers a global coverage but with a low repeat mission period. Global NWP models suffer from the same problems of precision and accuracy in marine coastal areas [9,10], although wind vectors are worldwide provided with good temporal and spatial resolutions (e.g., up to about 25 km resolution for the ECMWF Re-Analysis 5—ERA5 hourly reanalysis). They may not adequately reproduce the spatial variability of the local wind in such areas [11], considering that global estimates are given with a wind speed accuracy of about 2 m/s [12]. Regional NWP models are characterised by higher resolutions in time and space. They are optimised for lower scale dynamics, but they are unfortunately not always available.

Furthermore, synthetic aperture radar (SAR) systems have been demonstrated over the last decades to be a powerful instrument for observing the sea surface at high spatial resolution over large areas, hence providing wind speed and direction measurements over the ocean, including marine coastal areas as well. The importance of SAR-derived SSW retrievals is especially recognized, at both the global and local scale, in a wide range of applications such as marine meteorology (e.g., in the study of air-sea interaction [13] or hurricanes [14]), oceanography (e.g., regarding ocean dynamics research of wind-driven sea surface waves and currents [15]), and oil spill monitoring [16]. SAR applications are also used in the research on wind resource assessment [17] and environmental modelling [18].

The conventional way to report the accuracy of SAR SSW maps is through an overall error estimation by considering in situ or NWP models data used as reference. However, summary measures, such as the root mean square error (RMSE), the mean bias error (MBE), the square of the correlation coefficient ($R^2$), are not able to represent the spatial variation of the accuracy within the SSW maps obtained from retrieval algorithms. The spatial propagation of error is of great importance for modelling environmental processes, and ideally a map of an essential variable should be accompanied by a map of the spatial distribution of the related error. In other words, such a map would provide users a local estimation of accuracy.

As is well-known, a methodology for SSW speed estimation at high spatial resolution is represented by the so-called scatterometry-based approach, which relies on the exploitation of spaceborne SAR data through the inversion of a geophysical model function (GMF). The latter model stands as a semi-empirical relationship among the calibrated SAR amplitude, i.e., normalised radar cross section (NRCS) or alternatively named sigma nought ($\sigma_0$), the mean wind vector (speed and direction), and the mean radar incidence angle, all evaluated in a wind vector cell (WVC). A number of GMFs, some of them initially developed only for the C-band scatterometers [19–21], have been presented in the literature and also exploited by means of SAR images acquired at different frequency bands [22–30]. A particularly critical point is the requirement of SSW direction as a priori information in the GMF-based inversion scheme. A general way of obtaining such input direction is by the spatial resampling of the SSW direction provided from NWP model data [31], although they can show unpredictable errors in coastal areas [10]. SSW direction may also be inferred directly from wind-induced linear patterns (i.e., wind streaks, WSs [32], and boundary layer rolls, BLRs [33]) onto SAR amplitudes, and consequently SAR SSW speed may be derived at the same high spatial resolution, as well as at the same acquisition time of the SSW direction extracted from those SAR signatures.

Among the several methods for SSW direction extraction from SAR data [34–39], the local gradient-modified (LG-Mod) method [11] and its multi-scale (MS) version [40], developed by the same authors of this work, are suitable to locally provide not only the mean wind direction but also its related uncertainty within each examined WVC. The capability of the LG-Mod algorithm to furnish a spatial distribution of both SSW direction and the related uncertainty can be fruitfully exploited, whereas NWP models give all the

wind direction estimates with a constant maximum value of the uncertainty (e.g., 20° for ECMWF [12]). As shown in [11,40], each wind direction map can be accompanied by an associated uncertainty map. As a consequence, following the above-cited approach, the spatial distribution of both the SSW speed, along with the corresponding uncertainty, may be generated, in turn, in order to fulfil the further lack of wind accuracy information, especially near the coast.

The main aim of the study is to address an effective and simple operational methodology to fully evaluate the uncertainty in SAR SSW speed estimation when a GMF-based inversion is applied. The coupling between the unique ability of the LG-Mod algorithm to produce a spatial distribution of the SSW direction uncertainty and the methodology presented here can yield a spatial distribution of the SSW speed uncertainty as well. Moreover, focusing on the more recent C-band VV-polarised C-band model 5.N (CMOD5.N) [23] and C-band model 7 (CMOD7) [25] GMFs, a sensitivity analysis of the SSW speed uncertainty depending on both the uncertainties and the mean values of NRCS, incidence angle and SSW direction is carried out at different WVC dimensions (i.e., 5 km, 10 km, and 15 km).

A few authors have approached the problem concerning the uncertainty of SAR SSW speed inferred through a GMF inversion. In [12], SAR backscattering information is combined with some a priori information coming from high-resolution limited area models to retrieve the most probable wind vector, assuming that the NRCS, as well as the external wind (i.e., zonal U and meridional V wind components), contain errors and that these are well characterised in terms of Gaussian error standard deviations. In [41], the authors instead consider all sources of errors in the GMF inversion, and a sensitivity analysis of the wind speed uncertainty is proposed, referring to the C-band model 4 (CMOD4) [20] in particular. Ref. [42] presents a sensitivity study of wind speed retrieval with respect to the wind direction when using the X-band model 2 (XMOD2) GMFs developed at either the German Aerospace Center (DLR) or the Italian Space Agency (ASI). The study shows that an error of ± 10° for the SSW direction may introduce an error greater than 10% in the retrieval of SSW speed.

To the best of our knowledge, up to now no attempt has been made to provide spatial distribution maps of both SSW speed and direction uncertainty. These maps may be considered as an added value product with respect to the quality measurements provided by Copernicus. In fact, the SAR-derived wind field map available through Copernicus relies on statistical Bayesian inversion whose a priori information is obtained from NWP models [43]. The resulting map is provided with a wind quality flag based on the combination of two terms. The first one depends on the percentage of bright targets detected in the SAR image within the cell; the second one relates to the consistency between ancillary NWP model output and the NRCS used in the wind inversion scheme. The lower the minimum value of the cost function defined in the Bayesian procedure, the better the consistency and the higher the confidence level of the inverted wind vector must be [43]. However, as recalled previously, NWP models may fail to adequately retrieve SSW in complex coastal areas [10]. This may affect the quality flag provided by Copernicus. On the contrary, the definition of the SSW direction uncertainty provided by the MS LG-Mod depends only on the directional content evaluated from the NRCS in the WVC after the masking of unusable points [40], and then independently from any external NWP model reliability in such areas. As described in the paper, the SSW speed uncertainty does not depend on any external wind data source as well.

The remainder of the paper is structured as follows. Section 2 reports the available datasets information, while Section 3 describes in detail the proposed methodology and the comparison procedure. Section 4 shows and critically discusses results achieved in the estimation of the SAR SSW speed uncertainty, also examining the contribution of different parameters to the uncertainty and using NWP model data for some comparison. Finally, Section 5 summarises the conclusions.

## 2. Datasets

### 2.1. SAR Data

A number of 123 C-band Sentinel-1 (S-1) images, acquired by both A and B satellites, VV-polarised, both ascending and descending, Level 1 products, i.e., Interferometric Wide Swath Ground Range Multi-Look Detected High-Resolution images (afterwards, IW), are downloaded from the Copernicus Open Access Hub.

The S-1 dataset spans from January 2015 to November 2020 and covers the Atlantic European northwest shelf area. Figure 1 shows the full-frame coverage of all selected images, which are five descending (i.e., S1A_6234_37_14, S1A_20205_8_16, S1B_5838_37_13, S1B_10709_8_13, S1B_11482_81_13) and eight ascending (i.e., S1B_3687_161_11, S1A_4010_88_16, S1A_16931_59_11, S1A_17558_161_15, S1A_20562_15_15, S1A_21933_161_1, S1B_2360_59_10, S1B_8616_15_22). The study area is chosen among different benchmark sites of interest for the funding project E-SHAPE—myEcosystem showcase, which will serve focal user groups such as research, environmental assessment, reporting and management by offering seamless access to consistently scaled environmental information from various sources (e.g., SAR-derived products, like SSW maps). Nevertheless, the developed technique is applicable independently from the selected site.

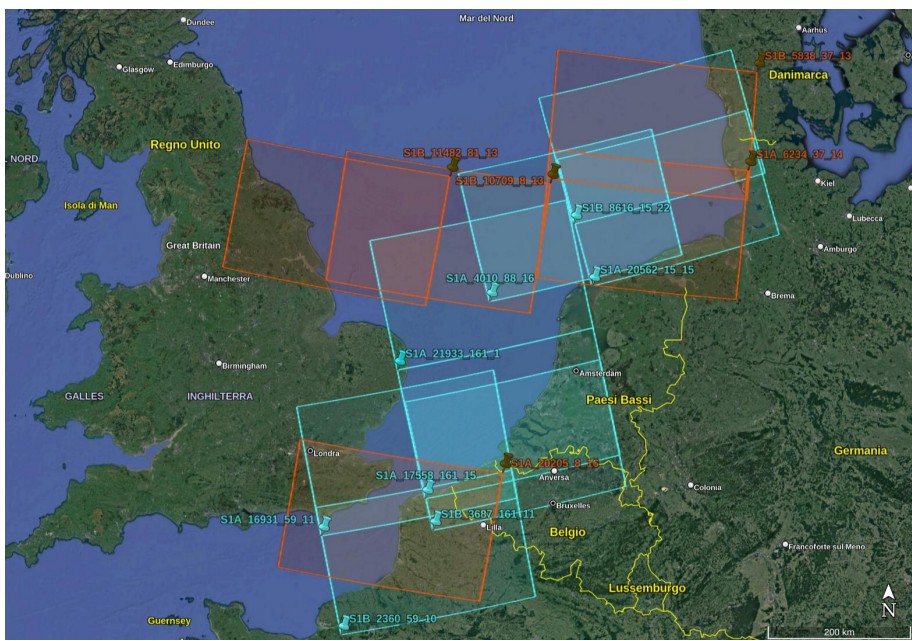

**Figure 1.** Atlantic European northwest shelf. Full-frame coverage is in orange and cyan for descending and ascending Sentinel-1 (S-1) Interferometric Wide Swath Ground Range Multi-Look Detected High-Resolution (IW) images, respectively. Background image (Data SIO, NOAA, US Navy, NGA, GEBCO; Image Landsat/Copernicus) is from Google Earth.

Those S-1 images characterised by a clear visibility (i.e., by human eye) of wind induced linear patterns, such as those from WSs [32], BLRs [33,44] or any other wind aligned patterns, hereafter "wind rows", are chosen aiming at extracting the local SSW direction exploiting such SAR signatures. In particular, the ultimate multi-scale version of the LG-Mod algorithm [40] is applied to serve this purpose.

The exploitation of C-band S-1 IW product type is to achieve the best MS LG-Mod performance in SAR derived SSW direction and uncertainty estimation (as experimented in [40]). Moreover, the choice of VV-polarised images is motivated to obtain the higher wind rows visibility, according to actual meteorological conditions, since SAR VV polarizations are slightly more sensitive than HH to wind pattern signatures [44].

### 2.2. NWP Model Data

Co-located 10-metre wind speed and direction are extracted from the global ERA5 reanalysis archive [45]. The archive makes available daily global re-analyses of wind data, every hour, and on regular latitude-longitude grids at $0.25° \times 0.25°$ resolution. The data are publicly available at: https://cds.climate.copernicus.eu (accessed on 20 February 2022) and they are here exploited only for a comparison of results.

## 3. Methods

### 3.1. Estimation Methodology for SAR SSW Speed and Its Uncertainty

A C-band VV-polarised GMF may be modelled as follows:

$$\sigma_0^{GMF}(\theta,W,\varphi) = B_0(W,\theta) \left[1 + B_1(W,\theta)\cos(\varphi) + B_2(W,\theta)\cos(2\varphi)\right]^n \tag{1}$$

where $\theta$, $W$ and $\varphi$ are the local magnitudes that represent respectively the SAR incidence angle, the SSW speed at the reference height of 10 m, and the azimuth angle (or SSW relative direction) between the SAR look direction and the SSW direction within a WVC. The coefficients $B_i$ (i = 0, ..., 2) are experimentally computed and tuned using a mixed dataset, typically comprising SAR measurements co-located with wind data from in situ stations, scatterometers and NWP models. The exponent n is 1.6 for the CMOD5.N [23]. Although the CMOD7 [25] was developed as the successor of the CMOD5.N, it does not have a closed-form expression as (1). The mismatch between the CMOD5.N and the backscatter measurements at low wind speeds was fixed in the CMOD7 by exploiting an independently developed ASCAT C-band GMF, namely C2013, which performs particularly well for low winds [46]. This yields that the CMOD7 is provided by tabled values of NRCS for a number of specific triplets $(\theta,W,\varphi)$.

Whatever the exploited GMF, SAR SSW speed W may be estimated from the minimization of a cost function J, defined as follows [42]:

$$J(W) = \left[\sigma_0 - \sigma_0^{GMF}(\theta,W,\varphi)\right]^2 \tag{2}$$

where $\sigma_0$, $\theta$ and $\varphi$ are the NRCS, the incidence angle and the SSW relative direction respectively, all measured as average values within a ROI. $\sigma_0^{GMF}$ represents the NRCS modelled by means of the GMF adopted for the SSW speed retrieval. From (2), the dependence of the function J on all the parameters may be explicitly expressed as $J(W) = J(\sigma_0,\theta,W,\varphi,\text{"GMF"})$.

Consequently, if the uncertainties in NRCS, incidence angle and SSW relative direction within the ROI, i.e., $\Delta\sigma_0$, $\Delta\theta$ and $\Delta\varphi$, respectively, are taken into account, the uncertainty in SAR SSW speed within the ROI itself may be experimentally estimated as $\Delta W = W' - W$, whereas W and W' are inferred from the inversion (minimization) problem in (2), and in the following expression (3), respectively:

$$J(W') = \left[(\sigma_0 + \varepsilon(\Delta\sigma_0)) - \sigma_0^{GMF}(\theta + \varepsilon(\Delta\theta),W',\varphi + \varepsilon(\Delta\varphi))\right]^2 \tag{3}$$

where $\varepsilon(\Delta\sigma_0)$, $\varepsilon(\Delta\theta)$ and $\varepsilon(\Delta\varphi)$ represent a generic error that affects the NRCS, the incidence angle and the SSW relative direction, and that occurs in the range $[-\Delta\sigma_0, \Delta\sigma_0]$, $[-\Delta\theta, \Delta\theta]$ and $[-\Delta\varphi, \Delta\varphi]$, respectively.

The SAR SSW speed uncertainty that corresponds to the generic error triplet $(\varepsilon(\Delta\sigma_0), \varepsilon(\Delta\theta),\varepsilon(\Delta\varphi))$ may be expressed by the notation:

$$\Delta W = \operatorname{argmin}_{W'}\{J(\sigma_0 + \varepsilon(\Delta\sigma_0),\theta + \varepsilon(\Delta\theta),W',\varphi + \varepsilon(\Delta\varphi),\text{"GMF"})\} - \operatorname{argmin}_W\{J(\sigma_0,\theta,W,\varphi,\text{"GMF"})\} \tag{4}$$

To obtain an estimation of the SAR SSW speed uncertainty, a number of configurations of the error triplet $(\varepsilon(\Delta\sigma_0),\varepsilon(\Delta\theta),\varepsilon(\Delta\varphi))$ $(\in \Sigma' \times \Theta' \times \Phi')$ need to be considered, provided that they are fully representative of the space $\Sigma \times \Theta \times \Phi$ of all possible error triples. It means that the examined configurations in $\Sigma' \times \Theta' \times \Phi'$ give the same maximum absolute

value of $\Delta W(\varepsilon(\Delta\sigma_0),\varepsilon(\Delta\theta),\varepsilon(\Delta\varphi))$ as the one obtained from the all the possible estimates in $\Sigma \times \Theta \times \Phi$. Hence, the maximum absolute value of the estimates $\Delta W$ derived from (4) for all the selected configurations is assumed as the SAR SSW speed uncertainty:

$$\Delta W_{max} = \max\{\left|\Delta W(\varepsilon(\Delta\sigma_0), \varepsilon(\Delta\theta), \varepsilon(\Delta\varphi))\right|, \text{ with}: (\varepsilon(\Delta\sigma_0), \varepsilon(\Delta\theta), \varepsilon(\Delta\varphi)) \in \Sigma' \times \Theta' \times \Phi'\} \qquad (5)$$

It is worth noting that the maximum uncertainty in SAR SSW speed, i.e., $\Delta W_{max} = f(\sigma_0, \Delta\sigma_0, \theta, \Delta\theta, \varphi, \Delta\varphi, \text{"GMF"}, \text{"J"})$, is affected by several parameters, which are:

i.　　The uncertainties $\Delta\sigma_0$, $\Delta\theta$ and $\Delta\varphi$. The two former are evaluated within each ROI as the standard deviation of the calibrated SAR backscatter (i.e., $\Delta\sigma_0 = \text{STD}(\sigma_0^{ROI})$) and the SAR incidence angle (i.e., $\Delta\theta = \text{STD}(\theta^{ROI})$), whereas the corresponding mean values are given by $\sigma_0$ and $\theta$, respectively. The latter uncertainty is instead represented by the error associated with the SSW direction $\varphi$, both estimated for the same ROI. The directional estimate $\beta^{ROI}$ (i.e., $\varphi = \beta^{ROI}$) and the marginal error $\text{ME}_\alpha^{ROI}$ (i.e., $\Delta\varphi = \text{ME}_\alpha^{ROI}$) are provided by the MS LG-Mod method, as described in [40].

ii.　　The average values $\sigma_0$, $\theta$ and the mean directional estimate $\varphi$. They are assumed as the true values of the corresponding parameters within the ROI, with $\sigma_0 = \text{MEAN}(\sigma_0^{ROI})$, $\theta = \text{MEAN}(\theta^{ROI})$ and $\varphi = \text{DIRMEAN}(\varphi^{ROI})$, respectively. The operator *DIRMEAN* computes the directional mean, thus assuming that directions $\varphi^{ROI}$ must be considered as axial data in the ROI [40]. The triplet $(\sigma_0, \theta, \varphi)$ represents the 'true state' for which the corresponding SAR SSW speed uncertainty is considered to be null.

iii.　　The specific GMF used to model the SAR sigma nought as a function of the SSW speed and direction, the SAR geometry, frequency and transmitting-receiving polarizations.

iv.　　The cost function J adopted for the minimization procedures. Without limiting the generality of the foregoing, a square function was chosen for this study as expressed by (2) and (3).

v.　　The approximation error for SAR SSW speed retrievals. In this work, the inversion procedure provides wind speed values in the range [0, 35] m/s with an approximation error of 0.1 m/s. The latter represents the step used in the iterative procedure to minimise the cost function.

Figure 2 illustrates the scheme of the methodology for the estimation of SAR SSW speed and its uncertainty, and the MS LG-Mod processing flowchart. The latter may provide not only the SSW direction and its uncertainty, but also SAR NRCS and the incidence angle, i.e., all SAR-derived input parameters for the proposed estimation methodology.

### 3.2. Comparison Procedure

SAR SSW estimates are derived from ROIs centred on the nodes of the ECMWF grid (originally of about 25 km by 25 km) resampled with a latitude/longitude spacing of about 5 km, 10 km and 15 km to match the corresponding ROI dimensions. No significant overlap is considered between adjacent ROIs. This means that the ROI dimension also represents the spatial resolution of the retrieved SSW. The average time delay between spatially co-located SAR estimates and ECMWF data is about 30 min, with the maximum value of 43 min.

ECMWF data are used as a reference for comparisons. Residuals estimated from SAR SSW speed and direction with respect to ECMWF ones, computed for each ROI, are compared with wind and direction uncertainty values obtained by the proposed method.

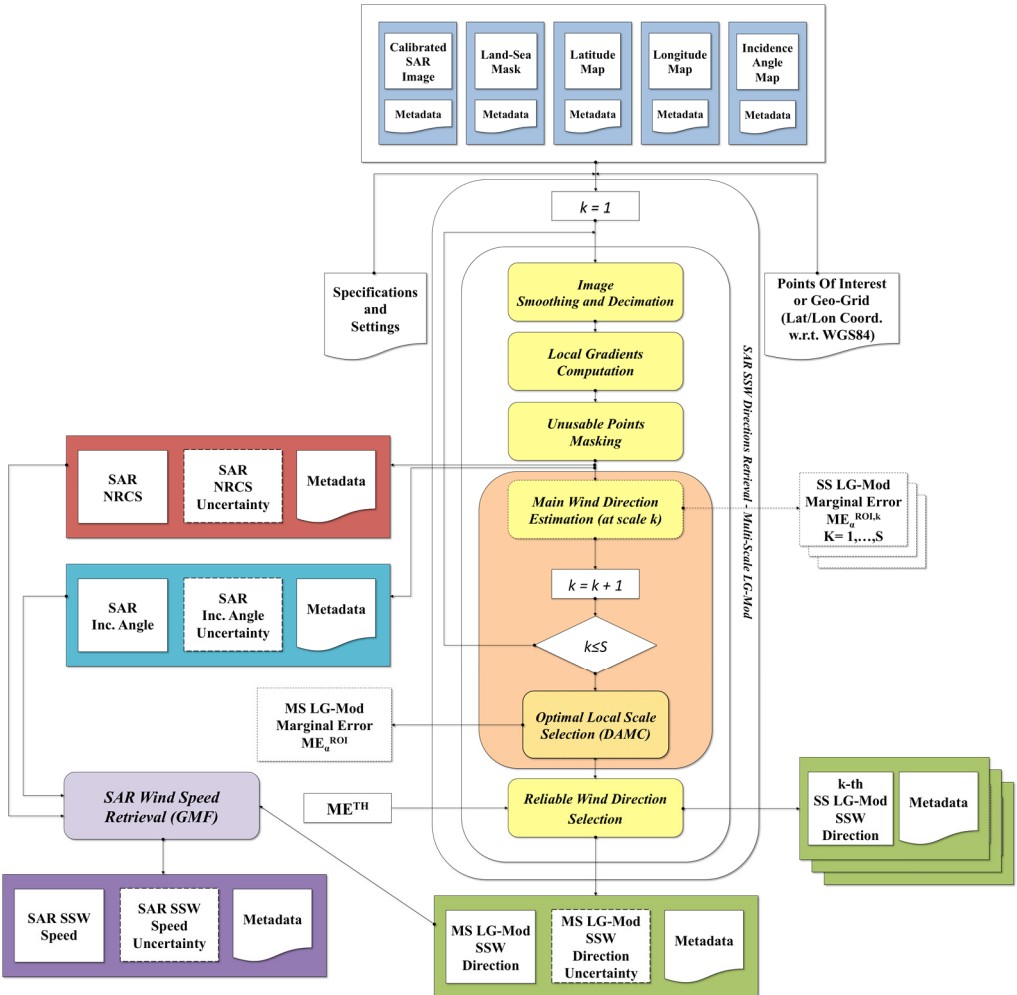

**Figure 2.** The scheme of the proposed methodology for the estimation of Synthetic Aperture Radar (SAR) Sea Surface Wind (SSW) speed and its uncertainty, and the Multi-Scale (MS) Local Gradient-Modified (LG-Mod) processing flowchart.

## 4. Results and Discussion

### 4.1. Statistics of SAR-Derived Parameters

Before reporting the results of the SSW speed uncertainty analysis from the above-described methodology applied to the available S-1 dataset, the SAR NRCS, incidence angle and SSW relative direction (the latter derived through the MS LG-Mod [40]) are analysed in terms of the statistical indicators of mean and standard deviation (or marginal error [40], in the case of the wind direction). These indicators are evaluated for all the ROIs enclosed within 65 km by 65 km sparse regions in the Atlantic European northwest shelf, where wind rows are clearly visible.

In order to investigate the effects of the considered spatial resolution of cells on the uncertainty in the SAR SSW speed estimation, different sizes of ROIs are considered. These ROIs are characterised by different square dimensions of about 5 km, 10 km and 15 km, providing a total number of useful cells (i.e., N_ROI) of 11,902, 2490 and 936, respectively.

#### 4.1.1. SAR NRCS, Incidence Angle and SSW Relative Direction

Table 1 collects helpful statistics at various ROI dimensions. In particular, mA_ROI$_i$ and sA_ROI$_i$ represent respectively the mean and the standard deviation of the generic SAR-derived input parameter A, which is, in turn, the NRCS, the incidence angle and the MS LG-Mod SSW relative direction, evaluated within the i-th ROI. The abbreviations min, max, mean and std stand for minimum, maximum, mean and standard deviation values

computed considering all the analysed N_ROI cells. For the actual S-1 dataset, tabled statistics suggest that:

A.　The MS LG-Mod SSW relative direction, expressed in the range [0°, 360°], is on average about 184°, with a small variation (growing) of standard deviation depending on the examined ROI dimension. This means that the visual selection of the S-1 regions characterised by clearly visible wind rows yielded to this prevailing wind direction. This fact confirms that, according to [44], wind rows detection depends strongly on the azimuth angle, with patterns detection rates at crosswind ($\varphi = 90°/270°$) being much lower (i.e., 3–10 times) than for upwind ($\varphi = 0°$) or downwind ($\varphi = 180°$). As regards the uncertainty associated with the SSW relative direction (by the MS LG-Mod), both the precision and the accuracy are decreasing (improving) with an increasing ROI dimension, ranging from 12.08° (at 5 km) to 4.16° (at 15 km) and from 5.56° (at 5 km) to 1.52° (at 15 km), respectively. As expected, a greater ROI size allows a better directional estimation of the wind, due to a higher number of usable points for the estimation within such dimension ROI [40].

B.　The incidence angle is quite stable around the average value of about 37.3°, with a small standard deviation which is basically independent from the ROI dimension. Hence, wind rows are most commonly observed at higher incidence angles in the available dataset, thus confirming again [44]. The uncertainty associated with the incidence angle increases (worsens) with an increasing ROI dimension in terms of both mean and standard deviation, ranging from 0.09° (at 5 km) to 0.27° (at 15 km) and from 0.01° (at 5 km) to 0.02° (at 15 km), respectively.

C.　The NRCS is about 0.1022 with a standard deviation of about 0.0526, considering the three different ROIs dimensions adopted for processing. The uncertainty of the NRCS slightly increases (worsens) with an increasing ROI dimension in terms of mean and standard deviation, varying from 0.0056 (at 5 km) to 0.0063 (at 15 km), and from 0.0029 (at 5 km) to 0.0035 (at 15 km), respectively.

**Table 1.** Statistics about the mean and standard deviation values of the input parameters SAR Normalised Radar Cross Section (NRCS), incidence angle and MS LG-Mod SSW relative direction for all Wind Vector Cells (WVCs or ROIs), at various dimensions.

| | NRCS | | | Incidence Angle (°) | | | MS LG-Mod SSW Relative Direction (°) | | |
|---|---|---|---|---|---|---|---|---|---|
| | 5 km | 10 km | 15 km | 5 km | 10 km | 15 km | 5 km | 10 km | 15 km |
| $\min_{\forall i,\ldots,N\_ROI}(\text{mA\_ROI}_i)$ [1] | 0.0165 | 0.0182 | 0.0178 | 31.76 | 32.05 | 32.12 | 74.42 | 92.66 | 81.34 |
| $\max_{\forall i,\ldots,N\_ROI}(\text{mA\_ROI}_i)$ [1] | 0.2887 | 0.2799 | 0.2770 | 45.03 | 44.80 | 44.68 | 282.25 | 269.86 | 267.86 |
| $\text{mean}_{\forall i,\ldots,N\_ROI}(\text{mA\_ROI}_i)$ [1] | 0.1020 | 0.1021 | 0.1024 | 37.34 | 37.35 | 37.33 | 184.11 | 184.04 | 184.07 |
| $\text{std}_{\forall i,\ldots,N\_ROI}(\text{mA\_ROI}_i)$ [1] | 0.0531 | 0.0525 | 0.0523 | 2.98 | 2.93 | 2.93 | 1.12 | 2.36 | 3.76 |
| $\min_{\forall i,\ldots,N\_ROI}(\text{sA\_ROI}_i)$ [1] | 0.0009 | 0.0007 | 0.0006 | 0.08 | 0.15 | 0.23 | 3.58 | 2.13 | 1.53 |
| $\max_{\forall i,\ldots,N\_ROI}(\text{sA\_ROI}_i)$ [1] | 0.0482 | 0.0318 | 0.0235 | 0.10 | 0.20 | 0.30 | 90.00 | 42.51 | 15.89 |
| $\text{mean}_{\forall i,\ldots,N\_ROI}(\text{sA\_ROI}_i)$ [1] | 0.0056 | 0.0055 | 0.0063 | 0.09 | 0.18 | 0.27 | 12.08 | 6.18 | 4.16 |
| $\text{std}_{\forall i,\ldots,N\_ROI}(\text{sA\_ROI}_i)$ [1] | 0.0029 | 0.0031 | 0.0035 | 0.01 | 0.01 | 0.02 | 5.56 | 2.40 | 1.52 |

[1] $\text{mA\_ROI}_i$ and $\text{sA\_ROI}_i$ represent respectively the mean and the standard deviation of the generic input parameter A (i.e., SAR NRCS, incidence angle and MS LG-Mod SSW relative direction) within the i-th ROI.

4.1.2. SAR SSW Speed and Its Uncertainty

Statistics about the SAR SSW speed (m/s) and the related maximum uncertainty (hereafter named simply uncertainty) (m/s) computed for all the ROIs, at the various dimensions and for the two GMFs, are collected in Table 2. From tabled values, it follows that:

a.　For each ROI size adopted, the two applied GMFs produce similar statistics, with an absolute difference of the mean and standard deviation of SSW speed of about 0.5 m/s and 0.3 m/s, respectively, and an absolute difference of the mean and standard deviation of the related uncertainty of about 0.1 m/s and 0.03 m/s or less,

respectively. All statistics obtained from the CMOD7 are smaller than those from the CMOD5.N. It appears that the CMOD7 allows better precision and accuracy in the estimation of SSW speed and its uncertainty. This is in accordance with the fact that the CMOD7 represents an improvement of the CMOD5.N in general, and especially at low winds [46].

b.   For each GMF applied, the mean (about 15.24 m/s and 14.77 m/s for CMOD5.N and CMOD7, respectively) and the standard deviation (about 3.53 m/s and 3.25 m/s for CMOD5.N and CMOD7, respectively) of SSW speed are quite independent from the ROI dimension. On the contrary, the mean and the standard deviation of SSW speed uncertainty significantly depend on the ROI size. In fact, the greater the ROI dimension, the lower the estimation statistics (i.e., between 2.07 m/s and 1.35 m/s for the mean, and 1.12 m/s and 0.58 m/s for the standard deviation, in the case of CMOD5.N; between 1.97 m/s and 1.26 m/s for the mean, and 1.11 m/s and 0.56 m/s for the standard deviation, in the case of CMOD7).

**Table 2.** Statistics about the SAR SSW speed (m/s) and the related uncertainty (m/s) for all ROIs, at various dimensions and for two different applied Geophysical Model Functions (GMFs).

| | CMOD5.N | | | CMOD7 | | |
|---|---|---|---|---|---|---|
| | **5 km** | **10 km** | **15 km** | **5 km** | **10 km** | **15 km** |
| $\min_{\forall i,\dots,N\_ROI}(W\_ROI_i)$ (m/s) [1] | 6.8 | 7.2 | 7.5 | 6.6 | 7.1 | 7.4 |
| $\max_{\forall i,\dots,N\_ROI}(W\_ROI_i)$ (m/s) [1] | 31.8 | 27.7 | 23 | 30.6 | 26.9 | 22.3 |
| $\text{mean}_{\forall i,\dots,N\_ROI}(W\_ROI_i)$ (m/s) [1] | 15.31 | 15.22 | 15.20 | 14.84 | 14.74 | 14.73 |
| $\text{std}_{\forall i,\dots,N\_ROI}(W\_ROI_i)$ (m/s) [1] | 3.64 | 3.49 | 3.46 | 3.37 | 3.21 | 3.18 |
| $\min_{\forall i,\dots,N\_ROI}(E\_ROI_i)$ (m/s) [1] | 0.4 | 0.4 | 0.5 | 0.4 | 0.4 | 0.4 |
| $\max_{\forall i,\dots,N\_ROI}(E\_ROI_i)$ (m/s) [1] | 12.7 | 4.7 | 4.4 | 12.2 | 4.8 | 4.2 |
| $\text{mean}_{\forall i,\dots,N\_ROI}(E\_ROI_i)$ (m/s) [1] | 2.07 | 1.39 | 1.35 | 1.97 | 1.31 | 1.26 |
| $\text{std}_{\forall i,\dots,N\_ROI}(E\_ROI_i)$ (m/s) [1] | 1.12 | 0.63 | 0.58 | 1.11 | 0.60 | 0.56 |

[1] $W\_ROI_i$ and $E\_ROI_i$ represent respectively the GMF-derived SSW speed (m/s) and the related uncertainty (m/s) within the i-th ROI.

Therefore, once a GMF is chosen, it is crucial to pay close attention to the balance between the desired spatial resolution of the SSW speed and the overall uncertainty achievable at such resolution, the latter uncertainty depending also on the actual SAR NRCS, incidence angle and SSW relative direction values, as well as their corresponding uncertainties (see Section 4.2). For example (Table 2), for our examined S-1 dataset, the use of the CMOD7 seems to provide on average a SSW speed uncertainty less than 2 m/s for all WVCs processed at 5 km, 10 km and 15 km. In detail, the number of WVCs that show an uncertainty less than 2 m/s is 6910 (of 11,902; 58.05%), 2119 (of 2490; 85.10%) and 811 (of 936; 86.65%) for 5 km, 10 km and 15 km processing, respectively.

Nevertheless, whatever the chosen GMF and the fixed WVC size, the proposed method allows the selection of only SSW speed values with an uncertainty lower than a user's tolerable limit. As a consequence, the method is defined by the powerful skill able to generate a spatial distribution map of SSW speed characterised anywhere by an uncertainty lower than a threshold value of acceptance (e.g., 2 m/s). This characteristic is similar to the one of the MS LG-Mod algorithms, which is able to discard those SSW directions that are considered not reliable enough when their uncertainty is higher than a threshold value of tolerance (e.g., 20°). On the latter point, the presented methodology may be used in an operational context as follows. Where wind-aligned patterns (e.g., wind rows) are visible on SAR images and local SAR-derived SSW directions are considered reliable within the MS LG-Mod approach, these directions and their uncertainties may be adopted as input to the methodology for SSW speeds and uncertainties estimation. Elsewhere, NWP model wind directions and their overall uncertainty may be exploited as well.

### 4.2. Contribution of Different Parameters to SAR SSW Speed Uncertainty

The uncertainty in SAR SSW speed obtained through a GMF inversion procedure depends on several parameters. In particular, the uncertainty $\Delta W_{max}$ is mainly affected by both the triplet of uncertainties ($\Delta\sigma_0,\Delta\theta,\Delta\varphi$) (bullet i, Section 3) and the true state triplet ($\sigma_0,\theta,\varphi$) (bullet ii, Section 3). Figures 3 and 4 highlight the relevance of all combining parameters.

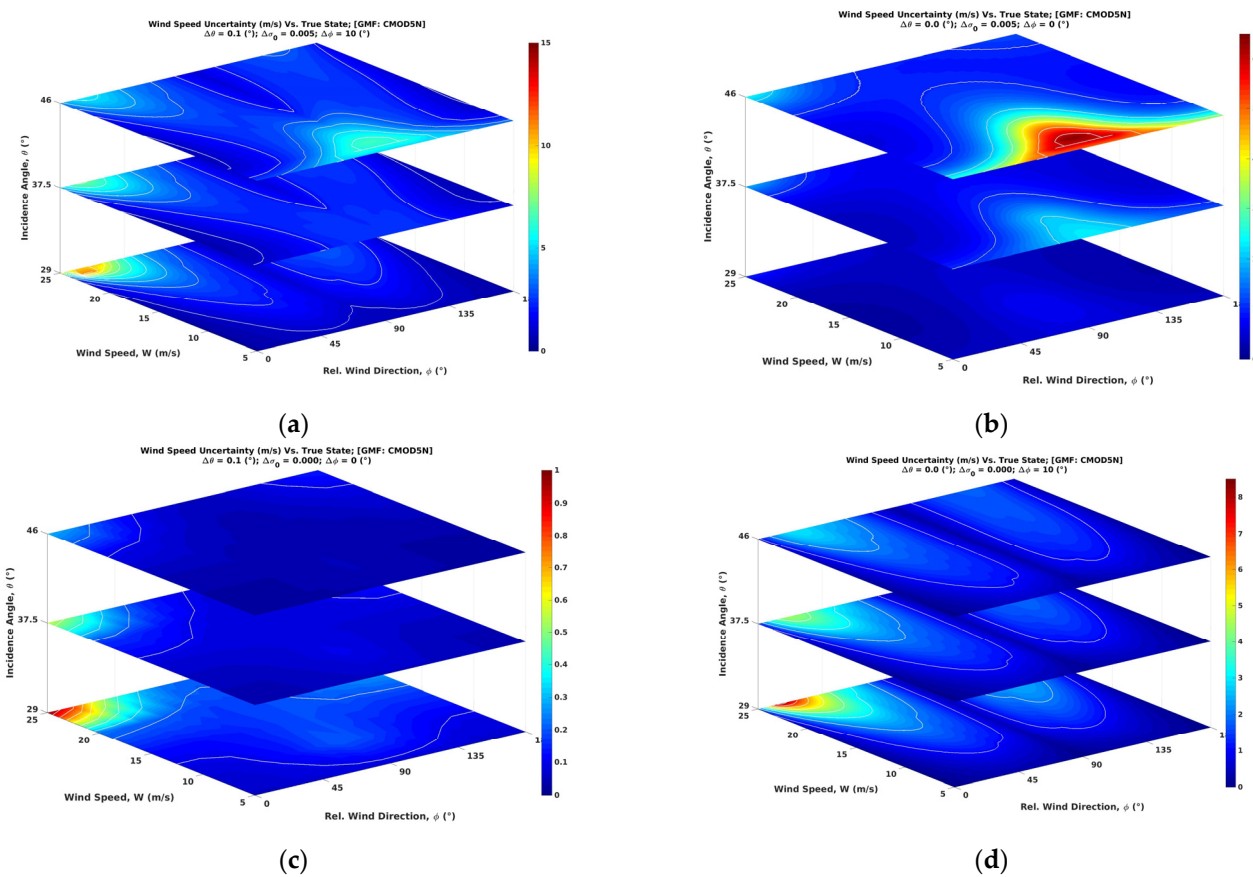

**Figure 3.** The total C-band MODel 5.N (CMOD5.N)-derived SAR SSW speed uncertainty $\Delta W_{max}$ (m/s), due to both the fixed triplet of uncertainties ($\Delta\sigma_0,\Delta\theta,\Delta\varphi$) = (0.005,0.1°,10°) and the varying true state ($W,\theta,\varphi$) (**a**). The contribution to this uncertainty only due to a single uncertainty among the ones regarding NRCS ($\Delta\sigma_0$ = 0.005) (**b**), incidence angle ($\Delta\theta$ = 0.1°) (**c**), and MS LG-Mod SSW (relative) direction ($\Delta\varphi$ = 10°) (**d**), with the same true state ($W,\theta,\varphi$). The wind speed uncertainty is computed for the wind direction $\varphi$ from 0° (upwind) to 180° (downwind) (*x* axis), the wind speed W between 5 and 25 m/s (*y* axis), and the incidence angle $\theta$ of 29°, 37.5°, and 46° (*z* axis).

Figure 3a shows, for a fixed triplet of uncertainties ($\Delta\sigma_0,\Delta\theta,\Delta\varphi$) = (0.005,0.1°,10°), the total CMOD5.N-derived SAR SSW speed uncertainty $\Delta W_{max}$ (m/s) plotted as a function of the varying true state ($W,\theta,\varphi$), where $\varphi$ is from 0° (upwind) to 180° (downwind) (*x* axis), W is between 5 and 25 m/s (*y* axis), and $\theta$ is 29°, 37.5°, and 46° (*z* axis). Note that the SSW speed W is reported on the *y* axis rather than the sigma nought $\sigma_0$ with the aim to allow for an intuitive understanding of the true state. Secondly, the SSW directions $\varphi$ from 180° to 360° are not plotted since the CMOD models, like CMOD5.N, are symmetric with respect to the direction $\varphi$ = 180°, once the wind speed W and the incidence angle $\theta$ have been fixed. Finally, the values on the *z* axis cover the range of incidence angles $\theta$ for Sentinel-1 acquisitions. The contribution to $\Delta W_{max}$ obtained considering each single uncertainty, i.e., the one derived from the NRCS ($\Delta\sigma_0$ = 0.005), the incidence angle ($\Delta\theta$ = 0.1°) and the MS LG-Mod SSW (relative) direction ($\Delta\varphi$ = 10°), with the same true state ($W,\theta,\varphi$), is shown in Figure 3b–d, respectively. From Figure 3b, the fixed uncertainty $\Delta\sigma_0$

weighs more for lower wind speeds, and especially for higher incidence angles. The wind direction that is more affected by this single uncertainty is the crosswind direction. When the only uncertainty $\Delta\theta$ is considered, as shown in Figure 3c, the SSW speed uncertainty decreases with increasing incidence angles and increasing wind speeds as well, with the upwind direction that represents the worst-case estimation. When the only uncertainty $\Delta\varphi$ is applied, Figure 3d shows again the decreasing SSW speed uncertainty with increasing incidence angles and wind speeds, with a worst-case estimation for a direction around the middle of the range [0°, 45°]. When these three contributions are all applied, their effects combine together in a complex way to provide the total SSW speed uncertainty, as reported in Figure 3a. The contribution derived from $\Delta\theta$ appears to be negligible with respect to those from both $\Delta\sigma_0$ and $\Delta\varphi$.

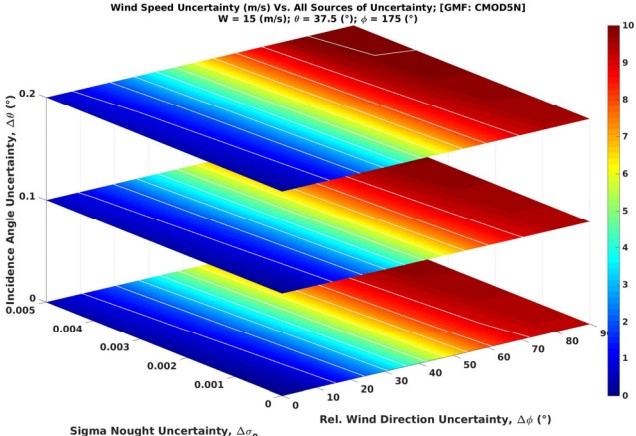

**Figure 4.** The CMOD5.N-derived SAR SSW speed uncertainty $\Delta W_{max}$ (m/s), due to both the fixed true state (W,θ,φ) = (15 m/s, 37.5°, 175°) and the varying triplet of uncertainties ($\Delta\sigma_0$,$\Delta\theta$,$\Delta\varphi$), respectively in the range [0, 0.005] (*y* axis), [0°, 0.2°] (with a step of 0.1°) (*z* axis) and [0°, 90°] (*x* axis).

Figure 4 displays, for a fixed true state (W,θ,φ) = (15 m/s,37.5°,175°), the CMOD5.N-derived SAR SSW speed uncertainty $\Delta W_{max}$ (m/s) plotted as a function of the varying triplet of uncertainties ($\Delta\sigma_0$,$\Delta\theta$,$\Delta\varphi$), respectively in the range [0, 0.005] (*y* axis), [0°, 0.2°] (with a step of 0.1°) (*z* axis) and [0°, 90°] (*x* axis). In the shown example, $\Delta W_{max}$ significantly increases with an increasing SSW direction uncertainty $\Delta\varphi$, and it also slightly increases with both increasing incidence angle and sigma nought uncertainties, $\Delta\theta$ and $\Delta\sigma_0$, respectively.

SAR Dataset Results

Table 3 reports statistics (in (m/s)) about the contribution to the total SAR SSW speed uncertainty (m/s) (E = $\Delta W_{max}$), due to the true state ($\sigma_0$,θ,φ), along with a single uncertainty term among the ones regarding the NRCS ($\Delta\sigma_0$) (E1), the incidence angle ($\Delta\theta$) (E2) and the MS LG-Mod SSW relative direction ($\Delta\varphi$) (E3), evaluated for all the ROIs, at various dimensions and for two applied GMFs. The ratio of each non-additive contribution to the total SSW speed uncertainty and the total SSW speed uncertainty itself (i.e., 100 × Ei/E (%), i = 1, 2, 3) is also reported in the table.

It is interesting to note that:

a. For each ROI size adopted, the CMOD5.N and the CMOD7 produce a similar ratio, for both mean and standard deviation, and for each contribution (i.e., E1, E2 and E3).

b. For each GMF applied, the ratio of both mean and standard deviation for each contribution depends on the ROI dimension. In particular, focusing on the ratio of the mean values, with an increasing ROI dimension, both the contributions E1 and E2, strictly due to the NRCS ($\Delta\sigma_0$) and the incidence angle ($\Delta\theta$) uncertainty, respectively, increases; on the other hand, the contribution E3, strictly due to the MS LG-Mod SSW relative direction uncertainty ($\Delta\varphi$), decreases. For example (Table 3),

for our S-1 dataset, the results for the CMOD7 application show that the contribution E1 varies from 34.5% (at 5 km) to 51.0% (at 15 km), the contribution E2 changes from 6.9% (at 5 km) to 22.3% (at 15 km), and the contribution E3 decreases from 59.9% (at 5 km) to 29.2% (at 15 km).

Therefore, with an increasing ROI dimension, the terms E1 and E2 weigh more, while the term E3 counts less, exactly at the same time that the uncertainty of the NCRS and the incidence angle worsen, and the uncertainty of the wind direction becomes more precise and accurate (Table 1). However, the combination of these concurrent factors basically produces a better estimation of the SAR SSW speed uncertainty for a greater ROI dimension (Table 2).

**Table 3.** Statistics (in (m/s)) and ratio (in (%)) about the contribution, due to the true state ($\sigma_0,\theta,\varphi$) and a single uncertainty among the ones regarding the NRCS ($\Delta\sigma_0$) (E1), the incidence angle ($\Delta\theta$) (E2) and the MS LG-Mod SSW (relative) direction ($\Delta\varphi$) (E3), to the total SAR SSW speed uncertainty (m/s) for all ROIs, at various dimensions and for two different applied GMFs.

| | CMOD5.N | | | CMOD7 | | |
|---|---|---|---|---|---|---|
| | **5 km** | **10 km** | **15 km** | **5 km** | **10 km** | **15 km** |
| $\mathrm{mean}\forall_{i,\dots,N\_ROI}(\mathrm{E1\_ROI}_i)$ (m/s) [1] | 0.61 (34.63%) | 0.59 (45.43%) | 0.67 (50.66%) | 0.57 (34.52%) | 0.55 (45.23%) | 0.63 (51.03%) |
| $\mathrm{std}\forall_{i,\dots,N\_ROI}(\mathrm{E1\_ROI}_i)$ (m/s) [1] | 0.27 (14.63%) | 0.28 (13.78%) | 0.31 (12.08%) | 0.25 (14.89%) | 0.26 (14.02%) | 0.29 (12.39%) |
| $\mathrm{mean}\forall_{i,\dots,N\_ROI}(\mathrm{E2\_ROI}_i)$ (m/s) [1] | 0.11 (6.65%) | 0.19 (14.99%) | 0.28 (22.03%) | 0.11 (6.87%) | 0.18 (15.20%) | 0.26 (22.30%) |
| $\mathrm{std}\forall_{i,\dots,N\_ROI}(\mathrm{E2\_ROI}_i)$ (m/s) [1] | 0.04 (3.47%) | 0.08 (5.96%) | 0.11 (7.66%) | 0.04 (3.77%) | 0.08 (6.13%) | 0.11 (7.86%) |
| $\mathrm{mean}\forall_{i,\dots,N\_ROI}(\mathrm{E3\_ROI}_i)$ (m/s) [1] | 1.34 (59.81%) | 0.62 (41.38%) | 0.42 (29.23%) | 1.29 (59.91%) | 0.59 (41.63%) | 0.39 (29.18%) |
| $\mathrm{std}\forall_{i,\dots,N\_ROI}(\mathrm{E3\_ROI}_i)$ (m/s) [1] | 0.96 (15.26%) | 0.41 (14.68%) | 0.27 (12.49%) | 0.96 (15.26%) | 0.39 (14.86%) | 0.26 (12.63%) |

[1] $\mathrm{Ek\_ROI}_i$ represents the GMF-derived SSW speed uncertainty (m/s) within the i-th ROI, where k = 1, k = 2 and k = 3 means ($\sigma_0,\theta,\varphi,\Delta\sigma_0$), ($\sigma_0,\theta,\varphi,\Delta\theta$) and ($\sigma_0,\theta,\varphi,\Delta\varphi$), respectively.

Independently from the ROI dimension, both E1 and E3 are particularly significant, and consequently the uncertainty of both NRCS ($\Delta\sigma_0$) and the SSW direction ($\Delta\varphi$) must be taken into account in the estimation of the SAR SSW speed uncertainty. On the contrary, E2 and the effect of incidence angle uncertainty ($\Delta\theta$) may be ignored, especially for smaller ROI dimensions.

The previous results can be visualised in Figure 5 for the case of 5 km ROI dimension and the CMOD5.N GMF. Figure 5a plots in black the total amount of CMOD5.N-derived SAR SSW speed uncertainty derived considering the three uncertainties ($\Delta\sigma_0,\Delta\theta,\Delta\varphi$) and the true state ($\sigma_0,\theta,\varphi$) within each ROI. The value of SSW speed uncertainty components is obtained instead from the true state itself and, in turn, a single uncertainty term among the ones regarding the NRCS, incidence angle and MS LG-Mod SSW relative direction are reported in green, blue and red, respectively. Figure 5b shows, with the same colours, the ratio of each contribution with the total SSW speed uncertainty (i.e., $100 \times \mathrm{Ei}/\mathrm{E}$ (%), i = 1, 2, 3), for the same ROIs and the CMOD5.N. From Figure 5a, the higher contribution (mean = 1.34 m/s; std = 0.96 m/s, Table 3) seems clearly to derive from the uncertainty of the SSW direction, while the one (mean = 0.61 m/s; std = 0.27 m/s, Table 3) associated with the NRCS uncertainty is demonstrated to be meaningful as well. The lower contribution (mean = 0.11 m/s; std = 0.04 m/s, Table 3) is linked to the uncertainty of the incidence angle. These contributions, which depend also on the actual true state, shown in Figure 5b, represent on average about the 59.81% ($\pm$ 15.26%), 34.63% ($\pm$ 14.63%) and 6.65% ($\pm$ 3.47%)

of the total SSW speed uncertainty, as reported previously in Table 3. Looking at the trends in Figure 5b, an inverse correlation between the two dominant contributions due to the SSW direction and the NRCS uncertainty could be noticed. The reason could be that, when the directional content within a ROI rises, the SSW direction uncertainty decreases, and consequently the corresponding contribution to the total SAR SSW speed uncertainty decreases as well. At the same time, a growing ROI directional content corresponds to an increase of the NRCS uncertainty, together with its contribution to the total SAR SSW speed uncertainty itself.

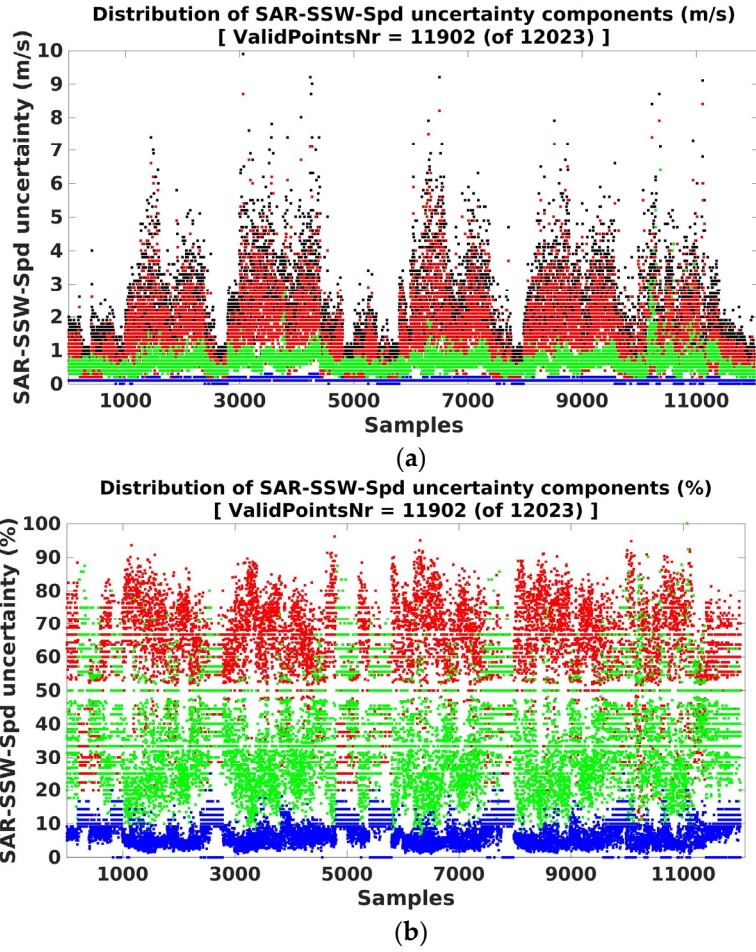

**Figure 5.** The total CMOD5.N-derived SAR SSW speed uncertainty $\Delta W_{max}$ (m/s) (purple), derived considering the three uncertainties ($\Delta\sigma_0, \Delta\theta, \Delta\varphi$) and the true state ($\sigma_0, \theta, \varphi$), and the contribution to this uncertainty due to the true state ($\sigma_0, \theta, \varphi$) and a single uncertainty among the ones regarding the NRCS ($\Delta\sigma_0$) (green), the incidence angle ($\Delta\theta$) (blue) and the MS LG-Mod SSW (relative) direction ($\Delta\varphi$) (red) (**a**). The corresponding ratio with reference to the total SSW speed uncertainty shown, in the range [0, 100] (%), with the same colours (**b**). Values are obtained for all the 5 km-ROIs.

Figure 6 shows, as sample spatial distributions, the NRCS $\sigma_0$ (a), the incidence angle $\theta$ (°) (c), the MS LG-Mod SSW direction $\varphi$ (°) (e), and the CMOD5.N-derived SSW speed $W$ (m/s) (g), with their related uncertainties $\Delta\sigma_0$ (b), $\Delta\theta$ (°) (d), $\Delta\varphi$ (°) (f) and $\Delta W_{max}$ (m/s) (h), respectively. All maps are obtained at 5 km by 5 km and refer to a 65 km by 65 km region in the Atlantic European northwest shelf, which was cropped from the VV-polarised, ascending, Sentinel-1 IW image acquired on 16th May 2018 at 17:33:28 UTC. Estimated SSW vectors are superimposed in each image of the figure. SSW directions derive from MS LG-Mod, while speeds are from the proposed methodology. SSW vectors are referred to the range-azimuth geometry using the true heading parameter of the S-1 image. According to average results obtained for the CMOD5.N and the 5 km ROI dimension

(Table 3, first column), the contribution $\Delta W_3$ due to the SSW direction uncertainty ($\Delta \varphi$, Figure 6f) is the one that weighs the most, for such conditions, in the total SAR SSW speed uncertainty ($\Delta W_{max}$, Figure 6h). The spatial variability in Figure 6f and h can suggest a kind of correlation between $\Delta \varphi$ and $\Delta W_{max}$, respectively.

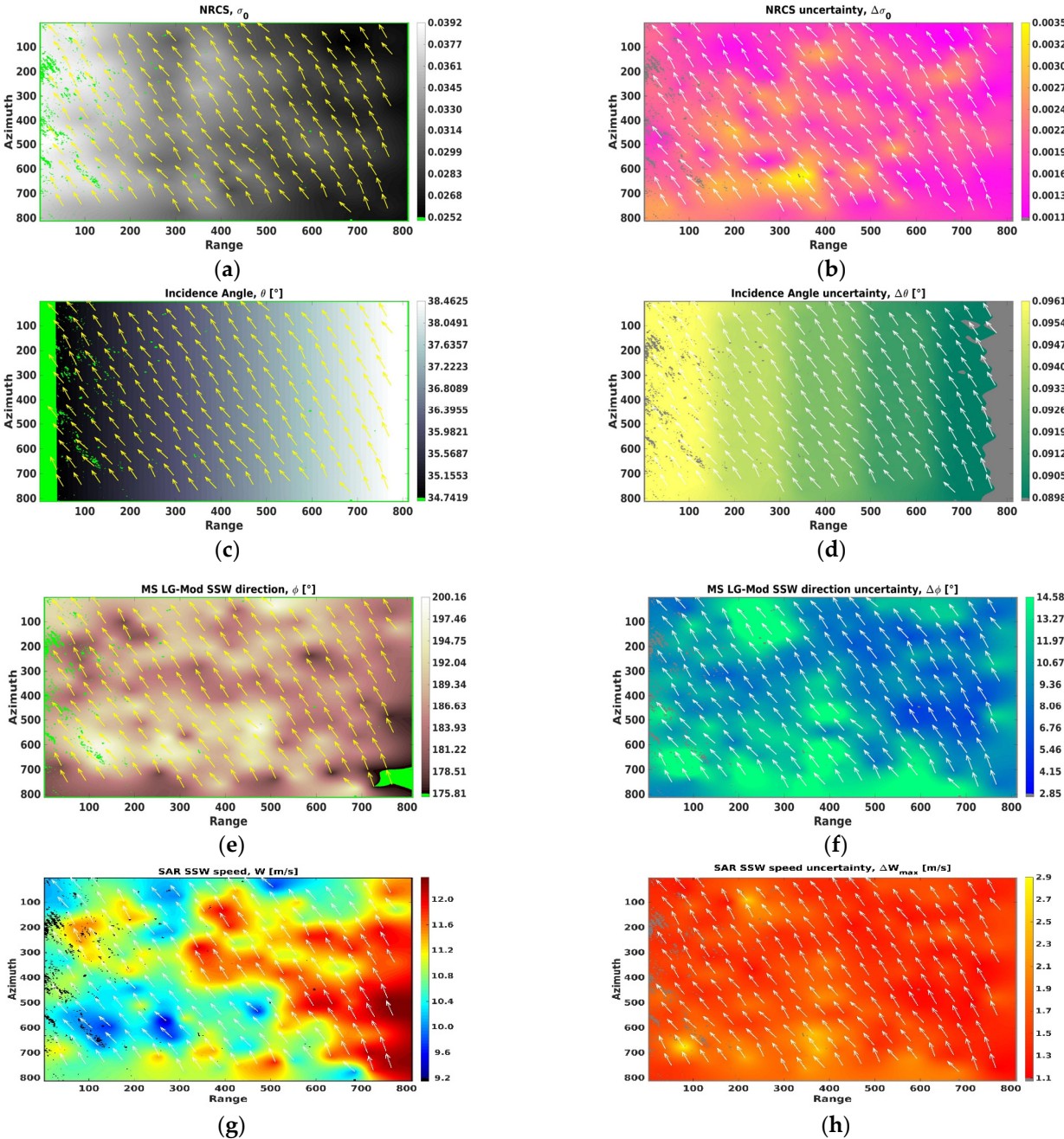

**Figure 6.** SAR NRCS $\sigma_0$ (**a**), incidence angle $\theta$ (°) (**c**), MS LG-Mod SSW direction $\varphi$ (°) (**e**) and CMOD5.N-derived SSW speed W (m/s) (**g**), with their uncertainties $\Delta \sigma_0$ (**b**), $\Delta \theta$ (°) (**d**), $\Delta \varphi$ (°) (**f**) and $\Delta W_{max}$ (m/s) (**h**), respectively. All maps are obtained at 5 km by 5 km from a Sentinel-1 IW image, ascending, VV-polarised, acquired on 16th May 2018 at 17:33:28 UTC. Estimated SSW vectors are superimposed.

Figure 7 illustrates the contribution to the total CMOD5.N-derived SAR SSW speed uncertainty $\Delta W_{max}$ (m/s) (Figure 6h), due to the true state ($\sigma_0, \theta, \varphi$), and only a single uncertainty term among the ones regarding the NRCS ($\Delta \sigma_0$) (a), the incidence angle ($\Delta \theta$)

(b), and the MS LG-Mod SSW (relative) direction ($\Delta\varphi$) (c). Again, all maps are obtained at 5 km by 5 km from the same S-1 IW crop image used for Figure 6. According to results shown in Figure 5, as well as to statistics reported in the first column of Table 3 (regarding the case of CMOD5.N and the 5 km ROI dimension), Figure 7a–c confirm respectively that the higher contribution to the total SSW speed uncertainty derives from the uncertainty of the SSW direction, the intermediate one is associated with the NRCS uncertainty, while the lower one is linked to the uncertainty of the incidence angle. The correlation between $\Delta W_{max}$ (Figure 6h) and $\Delta W_3$ (Figure 7c) is quite clear.

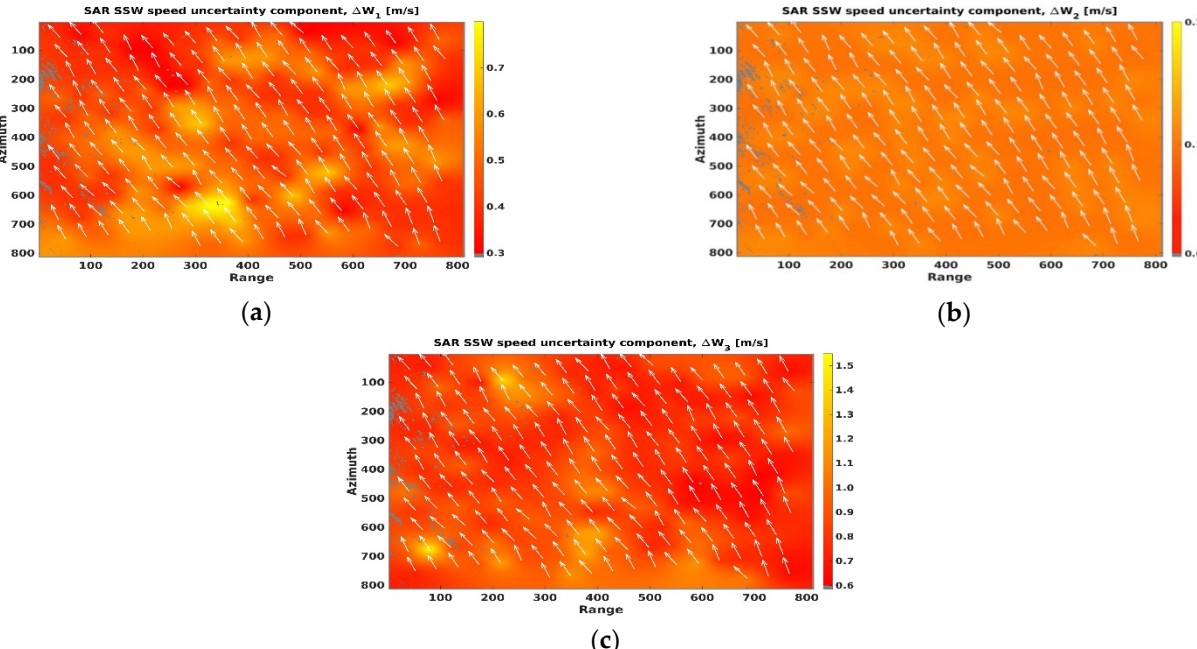

**Figure 7.** Contribution to the total CMOD5.N-derived SAR SSW speed uncertainty $\Delta W_{max}$ (m/s) (shown in Figure 6h), due to the true state ($\sigma_0,\theta,\varphi$) and a single uncertainty term among the ones regarding the NRCS ($\Delta\sigma_0$) (**a**), the incidence angle ($\Delta\theta$) (**b**), and the MS LG-Mod SSW (relative) direction ($\Delta\varphi$) (**c**). All maps are obtained at 5 km by 5 km from a Sentinel-1 IW image, ascending, VV-polarised, acquired on 16th May 2018 at 17:33:28 UTC. Estimated SSW vectors are superimposed.

### 4.3. Assessment of SAR SSW Direction and Speed Uncertainty as Proxy of Accuracy

To evaluate the results of SAR-derived SSW direction and speed uncertainty, the former derived from the MS LG-Mod algorithm and the latter from the proposed methodology and here applied to the S-1 dataset, wind data from the global ECMWF model were exploited. As all the examined ROIs are located on "open waters" in the Atlantic European northwest shelf, the ECMWF model data may be considered for comparisons.

The first comparison is made by means of SSW direction and speed values obtained by the MS LG-Mod and the GMF-based inversion (i.e., $\varphi$ and W, respectively) and the wind estimates from the ECMWF (i.e., $\varphi_{ECMWF}$ and $W_{ECMWF}$, respectively), thus providing summary statistics, such as the $R^2$, the RMSE and the MBE.

Figure 8a illustrates the scatterplot of the MS LG-Mod SSW direction (*y* axis) and the ECMWF model direction (*x* axis), for all the 5 km by 5 km ROIs (N_ROI = 11,902). The values of $R^2$ (=0.94), RMSE (=14.36°) and MBE (=5.10°) reveal a quite good agreement between the SAR-based technique (in the case of wind rows visibility) and the global NWP model (when applied to open waters) in wind direction estimation. Similar plots and results ($R^2$ = 0.97; RMSE = 11.53°; MBE = 5.19° and $R^2$ = 0.98; RMSE = 10.63°; MBE = 5.15°) were obtained for all the 10 km and 15 km square ROIs, respectively. Figure 8b reports instead the scatterplot of the SAR SSW speed (*y* axis) and the ECMWF model speed (*x* axis), for the same ROIs and the CMOD5.N. Statistics obtained from the comparison

between the SAR-derived SSW speed and the ECMWF one ($R^2$ = 0.53; RMSE = 3.20 m/s; MBE = 1.98 m/s, and $R^2$ = 0.53; RMSE = 2.76 m/s; MBE = 1.50 m/s for CMOD5.N and CMOD7, respectively) show less agreement in wind speed estimation compared to wind direction estimation, for both applied GMFs. This fact is true also for the 10 km ($R^2$ = 0.56; RMSE = 2.99 m/s; MBE = 1.88 m/s, and $R^2$ = 0.56; RMSE = 2.56 m/s; MBE = 1.41 m/s for CMOD5.N and CMOD7, respectively) and the 15 km ($R^2$ = 0.57; RMSE = 2.96 m/s; MBE = 1.89 m/s, and $R^2$ = 0.57; RMSE = 2.52 m/s; MBE = 1.41 m/s for CMOD5.N and CMOD7, respectively) processing results.

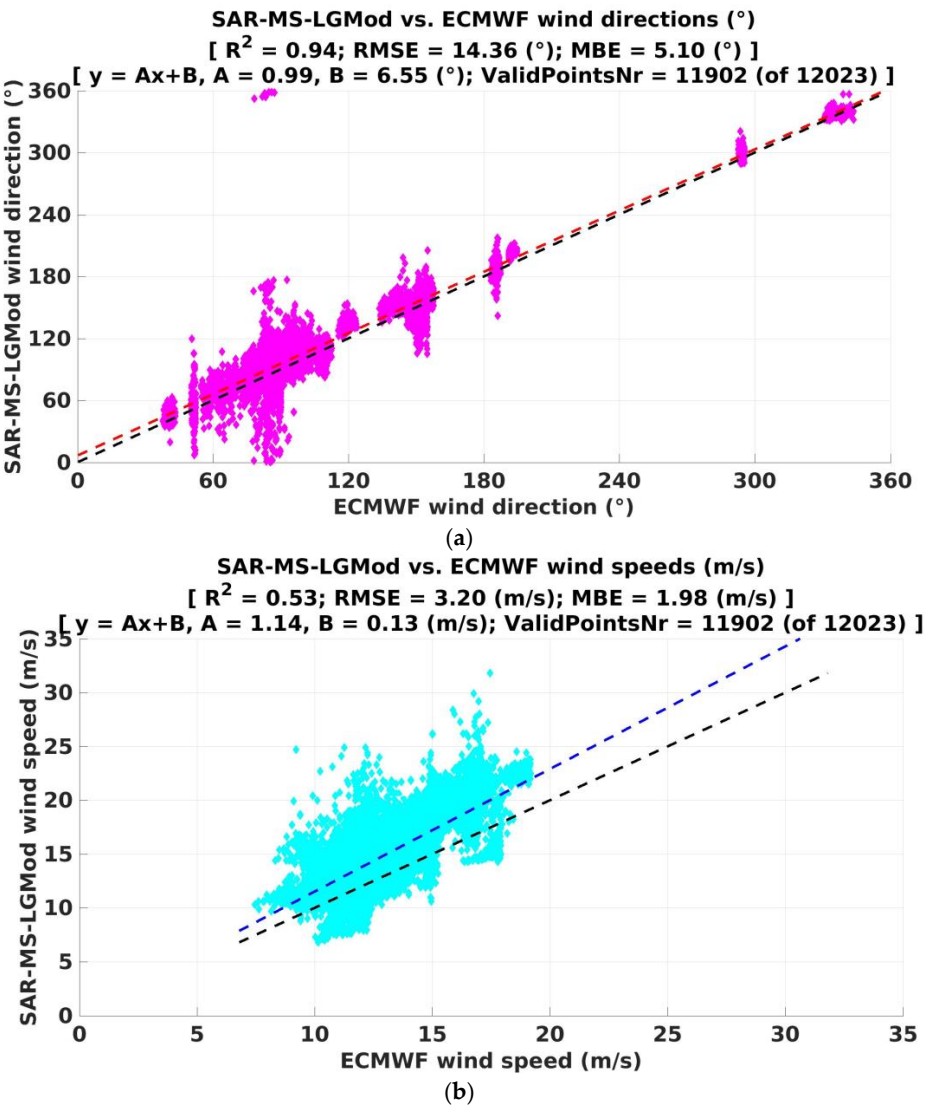

**Figure 8.** (**a**) Scatterplot of the MS LG-Mod SSW direction (*y* axis) and the European Centre for Medium-Range Weather Forecasts (ECMWF) model direction (*x* axis), for all the 5 km by 5 km ROIs (N_ROI = 11,902). Red dashed line shows the regression line with a slope A = 0.99. (**b**) Scatterplot of the SAR SSW speed (*y* axis) and the ECMWF model speed (*x* axis), for the same ROIs and the CMOD5.N. Cyan dashed line shows the regression line with a slope A = 1.14. Both black dashed lines represent the reference line with a slope of 1.

Another evaluation is based on the RMSE values computed as the root mean square of the SAR-derived SSW direction and speed uncertainty (i.e., $\Delta\varphi$ and $\Delta W_{max}$, respectively), on the one hand, and the absolute difference between the SAR-derived SSW direction and speed estimates and the ECMWF ones (i.e., $|\varphi\text{-}\varphi_{ECMWF}|$ and $|W\text{-}W_{ECMWF}|$, respectively), on the other hand.

Table 4 summaries these comparison results. From an overall point of view, there can be seen the same trends for the two different computations (namely, "SAR uncertainty" and "|SAR-ECMWF|" in the table), showing a decreasing (bettering) of RMSE values with an increasing ROI dimension. These findings are shared by estimations regarding both SSW direction and speed. However, the "SAR uncertainty" RMSE values exhibit an under-estimation with respect to the "|SAR-ECMWF|" evaluation.

**Table 4.** Comparison of Root Mean Square Error (RMSE): "SAR uncertainty" values derive from SAR SSW direction (°) and speed (m/s) uncertainty ($\Delta\varphi$ and $\Delta W_{max}$, respectively); "|SAR-ECMWF|" values refer to the difference between the SAR SSW direction and speed estimates and the ECMWF ones (($\varphi$-$\varphi_{ECMWF}$) and (W-W$_{ECMWF}$), respectively). Statistics obtained for all ROIs, at various dimensions. Two different GMFs applied for wind speed retrievals.

|  |  | MS LG-Mod SSW Direction (°) | | | CMOD5.N (m/s) | | | CMOD7 (m/s) | | |
|---|---|---|---|---|---|---|---|---|---|---|
|  | Based on | 5 km | 10 km | 15 km | 5 km | 10 km | 15 km | 5 km | 10 km | 15 km |
| RMSE | SAR uncertainty | 13.30 | 6.63 | 4.43 | 2.35 | 1.53 | 1.47 | 2.26 | 1.44 | 1.38 |
|  | |SAR–ECMWF| | 14.36 | 11.53 | 10.63 | 3.20 | 2.99 | 2.96 | 2.76 | 2.56 | 2.52 |

The final evaluation relies on the direct comparison between the SAR-derived SSW direction and speed uncertainty (i.e., $\Delta\varphi$ and $\Delta W_{max}$, respectively) and the absolute difference of the SAR-derived SSW direction and speed estimates with the corresponding ECMWF ones (i.e., |$\varphi$-$\varphi_{ECMWF}$| and |W-W$_{ECMWF}$|, respectively).

Figure 9 illustrates two binned scatterplots: (a) the MS LG-Mod SSW direction uncertainty, $\Delta\varphi$ (°), (*y* axis) plotted as a function of the absolute difference between the MS LG-Mod SSW direction and the ECMWF wind direction (*x* axis), for all the 5 km by 5 km ROIs (N_ROI = 11,902); and (b) the SAR SSW speed uncertainty, $\Delta W_{max}$ (m/s), (*y* axis) plotted as a function of the absolute difference between the SAR SSW speed and the ECMWF wind speed (*x* axis), for the same ROIs and the applied GMF CMOD5.N. For each bin (1 m/s and 5° for wind speed and direction uncertainty, respectively), the number of samples in the bin (N) and the aggregate statistics, m (mean) and s (standard deviation), are provided (unless N is too small, i.e., less than 1/100 of N_ROI).

Table 5 collects these further comparison results obtained for all ROIs, at various dimensions and with two different GMFs applied (for wind speed retrievals). The values of $R^2$ indicate a high correlation between the uncertainty derived from the two different computations, i.e., the one obtained only from SAR estimations and the other from the comparison SAR vs. ECMWF. The values of A (slope) and B (intercept) in a linear regression model show again (as in Table 4) an under-estimation of the SAR uncertainty with respect to the |SAR-ECMWF| uncertainty.

**Table 5.** Statistics derived from the direct comparison (scatterplot) between the SAR-derived SSW direction and speed uncertainty (i.e., $\Delta\varphi$ and $\Delta W_{max}$, respectively) and the absolute difference of the SAR-derived SSW direction and speed estimates with the corresponding ECMWF ones (i.e., |$\varphi$-$\varphi_{ECMWF}$| and |W-W$_{ECMWF}$|, respectively). Statistics obtained for all ROIs, at various dimensions. Two different GMFs applied for wind speed retrievals.

|  | MS LG-Mod SSW Direction (°) | | | CMOD5.N (m/s) | | | CMOD7 (m/s) | | |
|---|---|---|---|---|---|---|---|---|---|
|  | 5 km | 10 km | 15 km | 5 km | 10 km | 15 km | 5 km | 10 km | 15 km |
| $R^2$ | 0.96 | 0.94 | 0.96 | 0.97 | 0.99 | 0.91 | 0.96 | 0.95 | 0.98 |
| y = Ax + B |  |  |  |  |  |  |  |  |  |
| A | 0.15 | 0.07 | 0.03 | 0.31 | 0.14 | 0.11 | 0.35 | 0.14 | 0.10 |
| B | 10.36 | 5.53 | 3.93 | 1.26 | 1.04 | 1.08 | 1.18 | 1.01 | 1.06 |

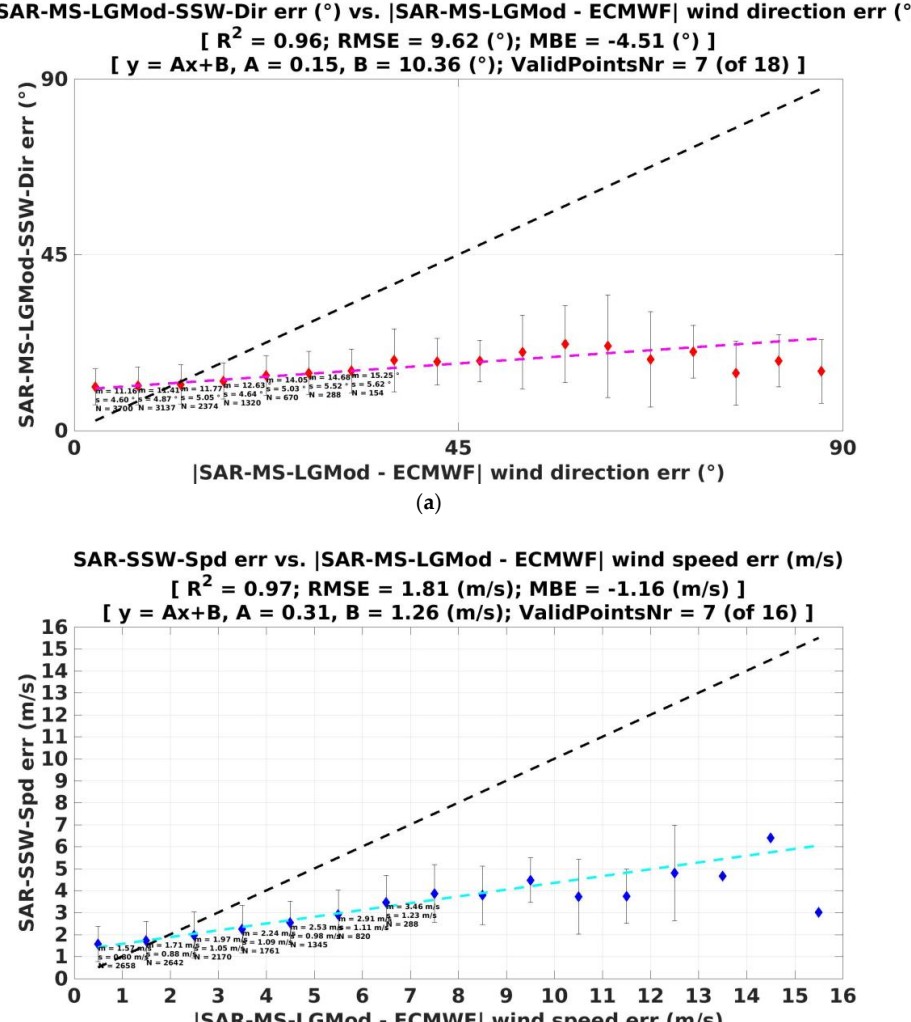

(a)

(b)

**Figure 9.** (**a**) 5°-binned scatterplot of the MS LG-Mod SSW direction uncertainty, Δφ (°), (*y* axis) vs. the absolute difference between the MS LG-Mod SSW direction and the ECMWF wind direction (*x* axis), for all the 5 km by 5 km ROIs (N_ROI = 11,902). Red dashed line shows the regression line with a slope A = 0.15. (**b**) 1 m/s-binned scatterplot of the SAR SSW speed uncertainty, $\Delta W_{max}$ (m/s), (*y* axis) vs. the absolute difference between the SAR SSW speed and the ECMWF wind speed (*x* axis), for the same ROIs and for the applied GMF CMOD5.N. Cyan dashed line shows the regression line with a slope A = 0.31. Both black dashed lines represent the reference line with a slope of 1. Note: For each bin, the number of samples in the bin (N) and the aggregate statistics, m (mean) and s (standard deviation), are provided (unless N is too small, i.e., less than 1/100 of N_ROI).

To achieve an accurate assessment of results, it should be recalled that:

- Although ERA5 wind data are gathered in this work in "open water" condition and assumed as reference for comparisons, these global NWP model data are doubtless affected by their own uncertainties, e.g., with a wind speed RMSE ranging from 1.25 to 1.5 m/s in the Dutch North Sea [47].
- SSW direction and the related uncertainty are provided by the MS LG-Mod algorithm as a directional mean and a measure of the directional content within each WVC [40]. However, the directionality of patterns in a cell may be sometimes affected by other local phenomena, even after a careful human eye selection of regions characterised by wind rows. In addition, as well known, the direction extracted by wind rows is not exactly aligned to the actual local wind direction [33], also depending on the phenomena causing the wind rows themselves.

- NRCS value and its uncertainty are assumed to be given by the mean and standard deviation of the NRCS in the WVC. It has been shown that the distribution of NRCS values is an important factor which can influence the accuracy of SAR wind and that should be taken into account [48]. Nevertheless, the sigma nought signal in the cell may depend on several factors other than the sea surface wind vector, which may be regarded as the dominant environmental factor responsible for the sea surface roughness. In fact, the effect of sea currents, wave motion, linear and non-linear waves in deep water and, to a lesser degree, bathymetry of shallow water, can represent other possible reasons for modification of the SAR backscattering [49]. Finally, error components derived from the calibration of the SAR signal might occur [12,43].
- GMFs applied represent a semi-empirical model whose coefficients are experimentally derived by tuning on a large amount of wind data, such as those from in situ stations, scatterometers and NWP models. Such tuning may determine further errors in the methodology results.

Last three bullets in the previous list represent general limits of application of the proposed methodology.

With these detailed observations, the resulting good correlation (Tables 4 and 5) however suggests that SAR-derived SSW direction and speed uncertainty, obtained from the MS LG-Mod and the proposed methodology, respectively, may be considered as proxies for the determination of the wind speed and direction accuracies. In principle, SAR SSW field maps and accuracy distribution maps may be supplied to models that require such wind information as input.

## 5. Conclusions

Following a scatterometry-based approach, which allows retrieval of SSW speed from SAR data through a GMF inversion, an effective and simple operational methodology has been addressed to fully evaluate the corresponding uncertainty. One of the key points of the presented methodology is precisely to provide a spatial distribution of SAR SSW speed uncertainty, moving beyond the limitations of the traditional approach to report the accuracy of SAR SSW speed in terms of summary measures, such as RMSE, MBE, and $R^2$. This result has been achieved exploiting the capability of the MS LG-Mod algorithm [40] to provide a spatial distribution of SSW direction and its uncertainty, both inferred directly from wind-induced linear patterns onto SAR amplitudes.

On the strength of this coupled approach, the estimation of the whole SSW field does not depend on any external wind data source, but only from SAR derived parameters. As a consequence, SAR SSW speed may be retrieved at the same high spatial resolution as well as at the same acquisition time of the SSW direction extracted from wind-driven SAR signatures. In case of wind rows visibility in SAR imagery, the use of low spatial resolution NWP models may be avoided, especially in complex coastal areas where they can show unpredictable errors [10]. This fact allows avoidance of temporal and spatial resampling of NWP model wind data, which is a further source of errors.

Hence, results obtained could mark an important step ahead towards the assessment of the suitability of high-resolution SAR data to fully describe local wind field spatial variability, also providing the spatial variations of the related wind speed and direction accuracy. Both accuracy maps are required by end users for the assimilation of satellite products, not only for modelling purposes but also for decision-making.

The sensitivity analysis of the GMF-inferred SSW speed uncertainty, which clearly depends on both the uncertainties and the mean values of the actual SAR NRCS, incidence angle and SSW direction, has been carried out at various WVC dimensions, i.e., 5 km, 10 km and 15 km. The more recent C-band VV-polarised GMFs, i.e., CMOD5.N and CMOD7, have been focused on always showing a better precision and accuracy of the latter model in the estimation of SSW speed and its uncertainty.

A noteworthy decrease (bettering) of SAR SSW speed uncertainty has been evidenced for an increasing WVC dimension, which is in contrast with the desired requirement of

using a small spatial resolution for wind retrievals. Nevertheless, whatever the chosen GMF and the fixed WVC size, the method is able to select only SSW speed estimates with an uncertainty that is lower than a suitable threshold of acceptance as defined by the user (e.g., 2 m/s). It follows that the user may also choose the smallest WVC dimension that guarantees a minimum desired percentage of SSW speed estimates with uncertainty lower than the fixed threshold. This is another strong point of the proposed methodology exploited in an operational context.

Independently from the WVC dimension and the applied GMF, both the contributions derived from the uncertainty of the NRCS and the SSW direction always appeared to be taken into account in the estimation of SAR SSW speed uncertainty. The contribution that may be ignored, especially for smaller WVC dimensions, is the one from the incidence angle uncertainty.

While considering the described limits of applicability of the approach, the resulting good correlation between the wind speed and direction uncertainties derived from SAR estimations and those from the comparison SAR vs. the global ECMWF, respectively, suggests that the proposed technique may be considered valid, although an under-estimation of the former estimates with respect to latter ones was observed.

Furthermore, it is crucial to highlight that the adopted methodology may be used in an operational context, exploiting MS LG-Mod SSW direction and uncertainty (where available), and a NWP model wind direction and its overall uncertainty (elsewhere) in a mixed approach.

Finally, in such context, the methodology may be applicable to any GMF and further SAR data bands (e.g., X-band) and polarizations (e.g., HV and VH), which will be, however, experimented in a future work. For instance, wind measurements under extreme weather conditions, such as hurricanes, could be fruitfully investigated within the scheme "MS LG-Mod algorithm + proposed methodology" by means of SAR cross-polarised images, thus avoiding saturation of VV polarised images [50,51]. Moreover, the application of the same scheme to marine coastal areas with heterogeneous and/or complex behaviours will be the focus in further experimentation.

**Author Contributions:** Conceptualization, F.M.R. and M.A.; methodology, F.M.R.; software, F.M.R.; validation, F.M.R. and M.A.; formal analysis, F.M.R.; investigation, F.M.R. and M.A.; resources, F.M.R. and M.A.; data curation, F.M.R. and M.A.; writing—original draft preparation, F.M.R.; writing—review and editing, F.M.R. and M.A.; visualization, F.M.R.; supervision, F.M.R. and M.A.; project administration, M.A.; funding acquisition, M.A. All authors have read and agreed to the published version of the manuscript.

**Funding:** This work was supported by the European Union's Horizon2020 research and innovation programme, within the project E-SHAPE—myEcosystem showcase, Grant Agreement ID 820852.

**Institutional Review Board Statement:** Not applicable.

**Informed Consent Statement:** Not applicable.

**Data Availability Statement:** Data sharing not yet available.

**Conflicts of Interest:** The authors declare no conflict of interest. The funders had no role in the design of the study; in the collection, analyses, or interpretation of data; in the writing of the manuscript, or in the decision to publish the results.

## Abbreviations

Acronyms in the manuscript are listed below:

| | |
|---|---|
| ASCAT | Advanced scatterometer |
| ASI | Agenzia Spaziale Italiana (Italian Space Agency) |
| BLR | Boundary layer roll |
| CMOD | C-band model |
| DIRMEAN | Directional mean |
| DLR | Deutsches Zentrum für Luft- und Raumfahrt (German Aerospace Center) |

| | |
|---|---|
| ECMWF | European Centre for Medium-Range Weather Forecasts |
| ECV | Essential climate variable |
| ERA5 | ECMWF Re-Analysis 5 |
| GCOS | Global Observing System for Climate |
| GMF | Geophysical model function |
| IPCC | Intergovernmental Panel on Climate Change |
| JASON | Joint Altimetry Satellite Oceanography Network |
| LG-Mod | Local gradient-modified |
| MBE | Mean bias error |
| ME | Marginal error |
| MS | Multi-scale |
| NDBC | National Data Buoy Center |
| NOAA | National Oceanic and Atmospheric Administration |
| NRCS | Normalised radar cross section |
| NWP | Numerical weather prediction |
| $R^2$ | Square correlation coefficient |
| RMSE | Root mean square error |
| ROI | Region of interest |
| S-1 | Sentinel-1 |
| SAR | Synthetic aperture radar |
| SSW | Sea surface wind |
| STD | Standard deviation |
| UNFCCC | United Nations Framework Convention on Climate Change |
| VV | Vertical transmitting vertical receiving |
| WRF | Weather research and forecasting |
| WS | Wind streak |
| WVC | Wind vector cell |
| XMOD | X-band model |

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
