# Peer review of "Uncertainty Analysis in SAR Sea Surface Wind Speed Retrieval through C-Band Geophysical Model Functions Inversion"

_remotesensing, doi:10.3390/rs14071685_

Round 1

Reviewer 1 Report

The problem addressed is of undeniable interest for scientists working in the field of oceanic remote sensing, in particular on the problem of estimating the wind vector from SAR radar images. This is a problem where there is a lot of work with the exploitation of radar images collected in C band but also in X band. Based on a set of SAR Sentinel-1 images collected in C band , the methodology adopted by the authors relates to the inversion of the GMFs with a study of the sensitivity of the uncertainty of the velocity vector. Also, an assessment was carried out considering the wind data from the model of the European Center for Medium-Range Weather Forecasts (ECMWF). The organization and structure of the paper is logical. The problem, the objectives and the methodology adopted are well presented. The results obtained and the analyzes carried out are convincing. However, it would be useful to consider the comments and questions given below.

- It would be useful to take into account other work carried out and published in the literature, a non-exhaustive list is given below.

- It would be important to specify why only the images collected in band C are used. Also, why not look at all the polarizations including the crossed polarizations.

- It would be important to specify the limits of the methodology adopted according to the area observed (coastal which can be heterogeneous and with complex behaviors, in deep water with linear and non-linear waves, etc.), the type of sensor, and the methods used for inversion taking into account their limit… and during complex situations with hurricanes…

- It would be useful to

- It would be useful to specify how to implement the adopted methodology in a practical and/or operational context.

- The conclusion should be strengthened and include some guidance for future work.

Reviewer 2 Report

The uncertainty of the sea surface wind (SSW) speed retrieved from SAR images was studied by uncertainties of SAR Normalized Radar Cross Section (NRCS), incidence angle and SSW direction. It is well understood that wind speed can be retrieved from SAR images while wind direction is generally hard to be retrieved unless under certain conditions, e.g., when there exist wind rows in images. So readers may expect a new method to derive wind directions from SAR images but it is truly a difficult problem. Methods and analysis are quite thorough and results are scientifically sound.

Problems:

  1. ERA5 dataset from ECMWF have been released for several years and have higher resolution for wind data, so, why did not you use this dataset to compare with SAR images with high resolution?
  2. The coordinates in Figures 5 and 6 are in range-azimuth without geo-projection, but the true SSW directions are superimposed onto these images (if this is TURE), so is it wrong to do so?

Now some suggestions:

  1. It can only be indicated from the title that only wind speed was studied, thus the title should be modified according to the contents.
  2. In titles of Figures 2a-2d, the unit of â–³sigma naught is wrong (at present m/s).

Reviewer 3 Report

[General comments]
This paper deals with the assessment of the suitability of Synthetic Aperture Radar (SAR) data for retrieving sea surface wind data at the region of interest (ROI). In my opinion, the scientific procedure meets the standard except that the details of the numerical model data used are not available. The presentation of the paper also meets the standard except some captions should be improved. The results obtained in this study are interesting.
However, many abbreviations are particularly difficult to understand for readers who are not familiar with this research field. I would like to recommend that a Table for the abbreviations be added in this paper. Therefore, I recommend that this manuscript be published after minor revision based on the following specific comments.

[Specific comments]
1. Please write down ROI, CMOD7, GMF, and CMOD5.N. in the abstract. In addition, please write down them when they first appear in the text.
2. Please provide a schematic diagram on the algorithm used in this study as shown in Fig. 3 of Rana et al. (2019). In the schematic diagram, please make sure that the focus points proposed in this study can be easily seen.
3. Please correct the typos in lines 270 and 560, and the color bar in Fig. 6 in line 491.
4. The regression line in Fig. 7 needs to be explained in the caption; there are two lines so that the reader may be confused at the first look. Please check other captions as well.
5. I would like the authors to explain the details of ECMWF data. How to interpret the 587 lines depends on whether the data set used is a reanalysis data set or a prediction data set: This is because the meaning of uncertainty is different between the two.

Reviewer 4 Report

This manuscript describes an effective and simple operational methodology to fully evaluate the uncertainty in SAR sea surface wind (SSW). Authors proposed the method using Geophysical Model Function (GMF) inversion for a spatial distribution of both SSW and its uncertainty. I think that this study is applicable to any GMF and further SAR data. I think it will be better manuscript if authors revise the following:

  1. First of all, this manuscript as a whole (especially Introduction Section) is so subdivided that it is quite difficult to understand the flow of content. Authors need to organize the contents again by grouping paragraphs by similar topics more easily for the reader to understand.
  2. In Section 2, authors should add more explanations about the study area. In addition, in Introduction or Section 2, it should be explained why this study is necessary in the current study area, not the general necessity.
  3. It is awkward that only Section 3.1 part is separated. It would be better to subdivide Section 3 and divide it into 3.1 and 3.2.
  4. If possible, it would be readable to separate Results and Discussion Sections.
  5. In addition, I think that it would be desirable to present the key points obtained through the study rather than to describe the summary of the results long.
